# GuirlVG: Incentivize GUI Visual Grounding via Empirical Exploration on Reinforcement Learning

**Weitai Kang**[1] **Bin Lei**[2] **Gaowen Liu**[3] **Caiwen Ding**[2] **Yan Yan**[1]
[1]University of Illinois Chicago  [2]University of Minnesota  [3]Cisco Research

## Abstract

Graphical user interface visual grounding (GUI-VG)—a core capability for GUI agents—has primarily relied on supervised fine-tuning (SFT) of multimodal large language models (MLLMs), demanding extensive data curation and significant training costs. However, as MLLMs continue to advance and even cover GUI domains during pretraining, the necessity of exhaustive SFT post-training becomes increasingly questionable. Meanwhile, the recent successes of rule-based reinforcement fine-tuning (RFT) suggest a more efficient alternative. However, despite its promise, the optimal manner of RFT for GUI-VG remains unexplored. To bridge this gap, we introduce *GuirlVG*, a reinforcement learning–based GUI-VG method built on a systematic empirical study and a novel stabilization technique. Preliminarily, we find that naive application of RFT underperforms the SFT baseline, motivating a deeper exploration of RFT. First, we decompose RFT into its core components and analyze the optimal formulation of each. Second, as part of this exploration, we propose a novel Adversarial KL Factor that dynamically stabilizes training to mitigate reward over-optimization. Third, we further explore the training configurations of RFT to enhance the effectiveness. Extensive experiments show that *GuirlVG*, with only 5.2K training samples, outperforms SFT methods trained on over 10M samples, achieving a **+7.7%** improvement on ScreenSpot, a **+17.2%** improvement on ScreenSpotPro and **91.9%** accuracy on ScreenSpotV2.

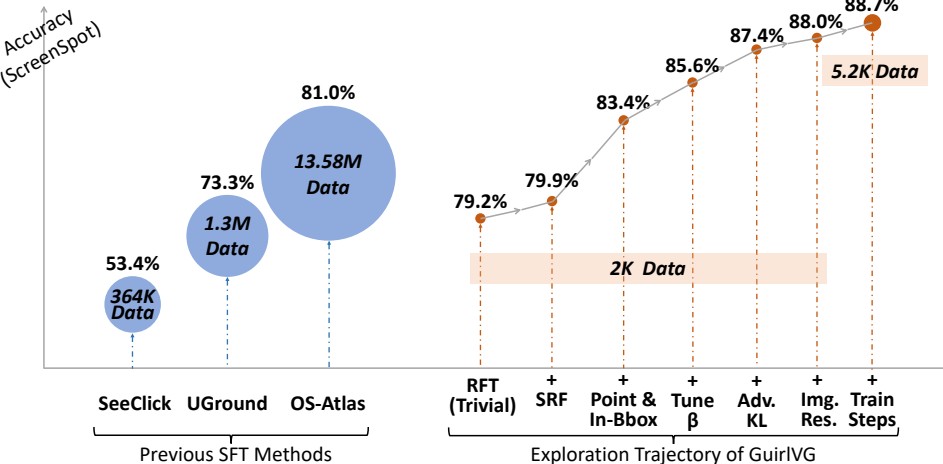

Figure 1: Step-by-step exploration of *GuirlVG*. Starting from trivial RFT, we progressively add Soft Reward Function, In-Bbox reward with point prediction, $\beta$ tuning, our Adversarial KL Factor, image resolution prompting, and extended training. With only 5.2K data, *GuirlVG* surpasses SFT methods trained on up to 13.58M data. Circle size reflects data scale used by each method.

## 1 Introduction

Graphical user interface (GUI) agents [Gou et al., 2024, Lin et al., 2024, Cheng et al., 2024, Qin et al., 2025, Xu et al., 2024, Huang et al., 2025b, Lei et al., Wu et al., 2024, Hong et al., 2024], empowered

by the rapid advancement of foundation models or multimodal large language models (MLLMs) [Liu et al., 2024b, Wang et al., 2024, Bai et al., 2025a], are increasingly capable of perceiving and acting within digital environments via screenshots. A core capability underpinning such agents is GUI visual grounding (GUI-VG)—the task of localizing actionable elements in a screenshot conditioned on a textual instruction [Gou et al., 2024, Cheng et al., 2024, Lei et al.]. Recent efforts have primarily approached GUI-VG through post-training of MLLMs via supervised fine-tuning (SFT), a paradigm that demands large-scale domain-specific data curation and significant training resources [Wu et al., 2024, Cheng et al., 2024, Qin et al., 2025, Gou et al., 2024, Lei et al.]. These advancements co-evolve with MLLM's capabilities, tailoring each generation of MLLMs to GUI-centric benchmarks.

However, SFT raises critical concerns regarding efficiency. As MLLMs continue to improve in general perception and reasoning—with some already ingesting GUI-related data during pretraining [Wang et al., 2024, Bai et al., 2025a]—the necessity of extensive post-training becomes increasingly questionable. Given the persistent training cost incurred with each new MLLM generation, a fundamental question arises: *Does exhaustive SFT remain the most effective post-training strategy?*

Meanwhile, the success of rule-based reinforcement fine-tuning (RFT) by Group Relative Policy Optimization (GRPO) [Shao et al., 2024a] in DeepSeek-R1 [Guo et al., 2025] inspires new directions. Recent methods transfer GRPO to visual grounding domains [Yuan et al., 2025, Luo et al., 2025, Shen et al., 2025, Bai et al., 2025b] with notable improvements. Despite these advances, no prior work has systematically studied RFT for GUI-VG. In fact, our results even reveal that naive application of RFT to GUI-VG under fair experimental settings underperforms the SFT baseline, prompting a critical question: *What is the optimal formulation of RFT objectives for GUI visual grounding?*

In this paper, instead of purely pursuing the best performance, *we focus on step-by-step and fair ablation to obtain rigorous findings that provide insights into how to design RFT for GUI-VG*. We do not compare with other RFT-based GUI-VG methods, since differences in data, training, and models would yield limited rigorous conclusions in systematic experiments. We elaborate on this point in section 2.1. We introduce *GuirlVG*, a RFT-based GUI-VG design built upon our comprehensive empirical study and a novel stabilization technique toward GRPO. ❶ We begin by deconstructing GRPO into its core components—format reward, accuracy reward, and KL penalty—and systematically ablate each component to derive an optimal configuration. ❷ To further address over-optimization caused by reward functions, we introduce a novel Adversarial KL Factor, which dynamically scales the KL penalty based on rewards to stabilize the learning process. ❸ Additionally, we explore a wide range of training setups, including hyperparameter tuning, LoRA enablement, and prompt engineering, to uncover best practices for effective RFT on GUI-VG. ❹ Finally, we conduct extensive experiments on ScreenSpot [Cheng et al., 2024], ScreenSpotV2 [Wu et al., 2024], and ScreenSpot-Pro [Li et al., 2025], demonstrating that *GuirlVG* achieves state-of-the-art results using as few as 2K~5.2K training examples. Compared to prior SFT baselines trained on hundreds of thousands to over ten million data, our method achieves superior accuracy with up to **+17.2%** absolute gains on ScreenSpotPro, highlighting the data efficiency and strong effectiveness of *GuirlVG*.

# 2 BACKGROUND

## 2.1 RELATED WORK

**Why do we need empirical studies?** While prior works [Yuan et al., 2025, Luo et al., 2025, Shen et al., 2025, Bai et al., 2025b] have proposed various modeling choices for RFT-based GUI-VG, these advances often emphasize reward function novelty or performance improvements without a systematic examination of underlying design factors. As GUI-VG continues to evolve rapidly, such one-off comparisons offer limited guidance for practitioners, since conclusions are often confounded by differences in data, training setups, and model structure. Empirical studies fill this gap by providing controlled and transparent analyses that isolate the effect of specific design choices. This type of investigation is essential for moving beyond ad-hoc innovation toward principled understanding, enabling the community to identify robust practices and avoid misleading interpretations of performance gains. There are some pioneer works for empirical studies in Multimodal Large Language Model, e.g. LLaVA-1.5 [Liu et al., 2024a], Prismatic [Karamcheti et al., 2024], Eagle [Shi et al., 2024] and Idefics2 [Laurençon et al., 2024]. However, there remains a lack of empirical studies investigating RFT in the context of GUI-VG.

**GUI Visual Grounding.** Visual grounding ability in graphical user interfaces [Cheng et al., 2024, Gou et al., 2024, Lei et al.] has become one of the main bottlenecks for AI agents [Gou et al., 2024, Lin et al., 2024, Cheng et al., 2024, Qin et al., 2025, Xu et al., 2024, Huang et al., 2025b, Lei et al., Wu et al., 2024, Hong et al., 2024]. To address this, SeeClick [Cheng et al., 2024] introduces a large-scale pretraining pipeline for GUI-VG and proposes an automated method to generate training data. Similarly, UGround [Gou et al., 2024] utilizes synthesized web-based data to support grounding training, and AGG [Lei et al.] builds a dedicated engine to collect extensive GUI images with annotations. OS-Atlas [Wu et al., 2024] further expands grounding data across multiple operating systems. UI-TARs [Qin et al., 2025] combines GUI-centric pretraining with task-conditioned fine-tuning to improve alignment between perception and reasoning. Despite the variety in their data construction, these methods commonly adopt the supervised fine-tuning (SFT) paradigm, which relies heavily on large volumes of high-quality labeled training data.

**Reinforcement Fine-Tuning.** Rule-based Reinforcement Fine-Tuning (RFT) with Group Relative Policy Optimization (GRPO) [Shao et al., 2024a] has recently demonstrated effectiveness across different domains [Shao et al., 2024b, Liu & Zhang, 2025, Wang* et al., 2025, Guo et al., 2025]. Unlike supervised fine-tuning (SFT), which enforces token-level supervision strictly corresponding to the answer, RFT encourages models to freely explore their reasoning process and provides supervision only at the level of the final output. This more flexible objective incentivizes stronger reasoning capabilities [Guo et al., 2025]. Furthermore, in the RFT algorithm—GRPO, task-specific rule-based reward functions are designed to provide supervision signals that are automatically verifiable. This eliminates the need for training a separate critic model [Schulman et al., 2017, Ouyang et al., 2022] or relying on human feedback [Kaufmann et al., 2023], thereby mitigating the risk of reward hacking [Weng, 2024] and making RFT an effective alternative to SFT.

## 2.2 PRELIMINARIES

**Group Relative Policy Optimization (GRPO).** Given a task input which additionally specifies a particular response format in the prompt, i.e. instructing the model to reason within $< think ><$ $think >$ tags and answer within $< answer >< /answer >$ tags, the model generates a group of $N$ candidate responses $\{o_1, o_2, \ldots, o_N\}$. Each candidate is evaluated using a rule-based reward function, yielding rewards $\{r_1, r_2, \ldots, r_N\}$. For each response $o_i$, this rule-based reward function scores two rewards: a format reward, $r_i^f$, which assesses whether the response adheres to the instructed tag structure, and an accuracy reward, $r_i^a$, which evaluates the correctness of the response, such as classification accuracy [Chen et al., 2025] or intersection-over-union (IoU) in detection tasks [Huang et al., 2025a, Liu et al., 2025b]. The total reward for is computed as

$$r_i = r_i^f + r_i^a. \tag{2.1}$$

The relative reward (also referred to as the advantage $A_i$) of the $i$-th candidate is computed by normalizing the rewards within the group of candidate responses:

$$A_i = \frac{r_i - \text{Mean}(\{r_1, r_2, \ldots, r_N\})}{\text{Std}(\{r_1, r_2, \ldots, r_N\})}, \tag{2.2}$$

where $\text{Mean}(\cdot)$ and $\text{Std}(\cdot)$ denote the mean and standard deviation, respectively. To stabilize training, GRPO additionally constrains model update by minimizing the KL divergence between the current model and a reference model (typically the original model). Thus, the objective $J_i$ to maximize for each candidate $o_i$ becomes

$$J_i = A_i - \beta \, \mathbb{D}_{\text{KL}}(o_i \, \| \, o_i^{\text{orig}}), \tag{2.3}$$

where $\beta$ is a hyperparameter controlling the KL penalty strength, and $o_i^{\text{orig}}$ is the corresponding response from the reference model. We omit details, such as clipping, averaging, etc.

**Implementation.** Unless specified otherwise, we fine-tune Qwen2.5-VL [Bai et al., 2025a] using LoRA [Hu et al., 2022] with a rank of 64 and an alpha of 128, while keeping the vision module frozen. Training data are randomly sampled from ShowUI [Lin et al., 2024], which crawls visually rich website data and augments desktop data from OmniAct [Kapoor et al., 2024] using GPT-4o [Hurst et al., 2024]. The group size of candidate responses, $N$, is set to 6, and the batch size is set to 4. The KL divergence coefficient ($\beta$) is set to 0.04 by default. The learning rate is set to $1 \times 10^{-5}$, with two

training epochs, AdamW optimizer, and a linear decay schedule. We use 6×NVIDIA A100-80G GPUs for training. For the SFT baseline, we adopt LLaMA Factory [Zheng et al., 2024] with the same training configurations for a fair comparison. For the efficiency and fairness of experiments, we report performances at step 500 for both RFT and SFT, where convergence is typically observed. Training beyond 500 steps yields only marginal improvements, with our final version reaching peak performance around step 1,300. Accordingly, our final version is only trained on 5,200 samples.

**Evaluation Suite.** We evaluate on three widely-used GUI-VG benchmarks across different platforms: ScreenSpot [Cheng et al., 2024], ScreenSpot v2 [Wu et al., 2024], and ScreenSpot-Pro [Li et al., 2025]. ScreenSpot evaluates GUI grounding capabilities across mobile, desktop, and web environments, while ScreenSpot v2 improves evaluation reliability by correcting annotation errors. ScreenSpot-Pro focuses on high-resolution professional scenarios, featuring expert-annotated tasks spanning 23 applications, five industries, and three operating systems. All benchmarks report the accuracy of whether the predicted point coordinate falls inside the ground truth bounding box of the corresponding element in the screenshot.

## 3 METHODOLOGY

### 3.1 CAN TRIVIAL ADOPTION OF RFT BEATS SFT?

We begin by comparing the SFT baseline with a trivial adoption of RFT for GUI-VG. Specifically, we adopt the commonly used implementation from HuggingFace [2025], Shen et al. [2025], using the following prompt for a given description of the target element:

*Please provide the bounding box coordinates [x1, y1, x2, y2] of a specific element based on this sentence: <description>. First, think through the reasoning process within <think> </think> tags. Then, output the bounding box coordinates in JSON format within <answer> </answer> tags.*

Table 1: Comparison of zero-shot, SFT, and trivial RFT on ScreenSpot (Qwen2.5-VL, 500 training steps).

| Method | Backbone | Step | Acc (%) |
|---|---|---|---|
| Zero-Shot | Qwen2.5-VL | 500 | 72.6 |
| SFT | Qwen2.5-VL | 500 | 82.6 |
| RFT (trivial) | Qwen2.5-VL | 500 | 79.2 |

For the format reward, a value of 1 is assigned if the output exactly matches the pattern "<think>...< /think>...<answer>...< /answer>", and 0 otherwise. The accuracy reward assigns 1 if a bounding box (bbox) array enclosed in a square bracket is detected and the IoU between the predicted and ground-truth bboxes exceeds 0.5, and 0 otherwise. During inference, the center of the predicted bbox is used as final prediction. Due to space limitations, we provide the detailed pseudo-code of RFT (trivial), along with the implementation details of the SFT baseline and the zero-shot setup for Qwen2.5-VL, in section A.1. As shown in table 1, both SFT and trivial RFT lead to improvements over the zero-shot baseline, but RFT (trivial) does not outperform SFT.

> **Finding 1.** Careful design of rule-based reinforcement fine-tuning, beyond common practice, is necessary for effectively improving GUI visual grounding performance.

### 3.2 HOW TO DESIGN REWARD FUNCTIONS IN GRPO?

As defined earlier, the default format reward enforces exact tag matching, while the accuracy reward relies strict JSON-style output consistent with the model's pretraining. The model is sharply penalized (rewarded 0) if any part of the expected structure is missing—such as an omitted </answer> tag—or it has a minor style deviation, e.g., outputting coordinates as a tuple instead of a JSON list. This rigid design introduces training noise and instability, even when the model successfully performs reasoning and answering.

To address this, we propose the Soft Reward Function (SRF), which provides partial credit to the presence of each tag and relaxes output style. Specifically, SRF removes the JSON requirement from the prompt. For the

Table 2: Compare the default reward function and our SRF on ScreenSpot (Qwen2.5-VL, 500 training steps).

| Method | Backbone | Step | Acc (%) |
|---|---|---|---|
| Default | Qwen2.5-VL | 500 | 79.2 |
| SRF (Ours) | Qwen2.5-VL | 500 | 79.9 |

format reward, SRF assigns +0.5 for each of `<think>` and `</think>`, +1/3 for each of `<answer>` and `</answer>`, and +1/3 if the content inside the answer tags contains the correct number of coordinates. The total score is normalized to [0, 1]. For the accuracy reward, SRF ignores style and simply extracts numeric values present in the output. Detailed prompts and pseudo-code are provided in section A.2 due to space constraints. As shown in table 2, SRF provides +0.7% improvement over the default reward function.

> **Finding 2.** Looser reward functions in RFT with fractional reward better support stable RFT training, and strict adherence to pretraining-style output is not necessary.

### 3.3 HOW TO DESIGN MODEL PREDICTION FORMAT ALONG WITH ITS ACCURACY REWARD FUNCTION?

The goal of GUI visual grounding is to predict a point that falls within the target element to enable the downstream action. To support this functionality, the most direct design is to predict a point and assign a binary reward based on whether it lies within the ground-truth bounding box [Shen et al., 2025] (In-Bbox). Alternatively, one can define reward based on a distance threshold $k$ [Liu et al., 2025a], where the point prediction is rewarded with 1 if it falls within $k$ pixels of the target center (denoted as Distance@$k$). Another option is to output a bounding box and derive a point prediction from its center, evaluating it via IoU with the ground truth. This can be used as a continuous reward or a thresholded one (e.g., IoU@0.5 gives a reward of 1 if IoU > 0.5, and 0 otherwise, as in our default format).

Table 3: Comparison of different prediction formats and accuracy reward functions under SRF on ScreenSpot (Qwen2.5-VL, 500 training steps).

| Prediction | Reward | Backbone | Step | Acc (%) |
|---|---|---|---|---|
| Bbox | IoU@0.5 | Qwen2.5-VL | 500 | 79.9 |
| Bbox | IoU | Qwen2.5-VL | 500 | 81.6 |
| Point | Distance@80 | Qwen2.5-VL | 500 | 82.7 |
| Point | In-Bbox | Qwen2.5-VL | 500 | 83.4 |

Building on our Soft Reward Function, we evaluate four configurations. The threshold of 80 for Distance@$k$ is empirically selected for best performance. As shown in table 3, Point prediction with In-Bbox performs best.

> **Finding 3.** The most effective RFT design is one that aligns directly with the task's functional goal— specifically, point prediction with in-bounding-box reward for GUI-VG.

### 3.4 HOW TO BALANCE THE KL PENALTY IN GRPO?

In GRPO, the KL penalty term enforces the current model to stay close to the original model, mitigating reward-driven over-optimization [Shao et al., 2024a, Guo et al., 2025]. The hyperparameter $\beta$ plays a critical role in determining the strength of this regularization. In our experiments, we observed that model performance is highly sensitive to this parameter.

We first empirically explore the effect of different values for $\beta$, then introduce a novel strategy we call *Adversarial KL Factor*, which dynamically scales the KL penalty based on reward strength. The intuition is that high-reward responses are more likely to cause over-optimization in GRPO. However, the KL penalty with the original model does not necessarily increase proportionally, especially when the original model itself assigns high probability to such responses. Therefore, a static KL term may fail to counterbalance the effect of reward. To address this, we define the *Adversarial KL Factor* as the ratio of the reward to its theoretical maximum $m$, and use it as a multiplicative modifier to $\beta$ to scale the KL penalty proportionally. This dynamic formulation ensures that as reward increases, the regularization also strengthens adaptively. The modified GRPO objective is:

$$J_i = A_i - \alpha_i \beta \, \mathbb{D}_{\text{KL}}(o_i \parallel o_i^{\text{orig}}), \quad A_i = \frac{r_i - \text{Mean}(\{r_1, r_2, \ldots, r_N\})}{\text{Std}(\{r_1, r_2, \ldots, r_N\})}, \quad \alpha_i = \frac{r_i}{m}, \qquad (3.1)$$

where $m = 2$ is the maximum possible reward under our setup.

Table 4: Comparison of different KL settings under SRF, point prediction, and In-Bbox reward on ScreenSpot (Qwen2.5-VL, 500 training steps).

| Adversarial | $\beta$ | Backbone | Step | Acc (%) |
|:---:|:---:|:---:|:---:|:---:|
| ✗ | 4e-2 | Qwen2.5-VL | 500 | 83.4 |
| ✗ | 0 | Qwen2.5-VL | 500 | 84.7 |
| ✗ | 1e-4 | Qwen2.5-VL | 500 | 85.6 |
| ✓ | 1e-4 | Qwen2.5-VL | 500 | 87.4 |
| ✓ | 1e-6 | Qwen2.5-VL | 500 | 77.5 |

Results are shown in table 4. Simply tuning $\beta$ provides clear performance improvements, demonstrating the importance of empirically calibrating the KL penalty. Notably, our *Adversarial KL Factor* strategy (row 4) achieves a substantial +1.8% gain over the best $\beta$ baseline (row 3), validating the advantage of dynamically adjusting KL strength in response to reward magnitude. Row 5 further indicates that setting $\beta$ too small results in degraded performance.

> **Finding 4.** GRPO is sensitive to the strength of the KL penalty, which requires empirical exploration. Our Adversarial KL Factor dynamically balances this penalty, leading to optimal performance.

## 3.5 SHOULD WE FULLY FINE-TUNE THE MODEL OR USE LORA?

We further investigate the impact of fine-tuning strategies by comparing full model fine-tuning (Full-FT) with LoRA [Hu et al., 2021] fine-tuning (LoRA-FT) applied to the LLM component. In practice, we observe that full fine-tuning tends to destabilize training unless a much smaller learning rate is used. Therefore, we reduce the learning rate for full fine-tuning to $1 \times 10^{-6}$, while keeping other hyperparameters consistent. We also report the training time per iteration using 6×A6000 GPUs.

Table 5: Comparison of Full-FT and LoRA-FT under SRF, point prediction, In-Bbox reward, $\beta = 1 \times 10^{-4}$, and Adversarial KL Factor on ScreenSpot (Qwen2.5-VL, 500 training steps). Training time is reported per iteration over 6×A6000 GPUs.

| Config | Backbone | Step | Time | Acc (%) |
|:---|:---|:---:|:---:|:---:|
| Full-FT | Qwen2.5-VL | 500 | 749.4 s | 87.5 |
| LoRA-FT | Qwen2.5-VL | 500 | 28.4 s | 87.4 |

As shown in table 5, Full-FT requires over 25 times more training time per iteration compared to LoRA-FT, while yielding only a marginal improvement of +0.1%. Given this modest performance gain relative to the substantial increase in computational cost, we adopt LoRA-FT as a more efficient strategy for GUI-VG reinforcement fine-tuning in our study.

> **Finding 5.** LoRA offers comparable performance to full fine-tuning while being significantly more efficient, making it a practical choice for GUI-VG with reinforcement fine-tuning.

## 3.6 HOW TO DECIDE THE GROUP SIZE AND BATCH SIZE IN GRPO?

The hyperparameters group size and batch size play critical roles in GRPO [Shao et al., 2024a]. Specifically, group size affects the normalization of advantage estimates, while batch size determines how each sample contributes to the final objective function. Therefore, it is necessary to empirically examine how different configurations of these two hyperparameters impact the final performance.

As shown in table 6, the configuration with group size 6 and batch size 4 achieves the highest accuracy, which is our default setting. Interestingly, increasing the group size from 6 to 8 leads to a substantial performance drop, even though larger groups theoretically provide better baseline estimates for advantage in GRPO to serve as a more stable substitute for the critic model in PPO [Schulman et al.,

2017]. This counterintuitive result suggests that RFT is sensitive to seemingly minor changes in implementation details and highlights the need for systematic validation of hyperparameter choices.

Table 6: Effect of group size and batch size under SRF, point prediction, In-Bbox reward, $\beta = 1 \times 10^{-4}$, Adversarial KL Factor and LoRA on ScreenSpot (Qwen2.5-VL, 500 training steps).

| Group | Batch | Backbone | Step | Acc (%) |
|---|---|---|---|---|
| 6 | 1 | Qwen2.5-VL | 500 | 86.5 |
| 6 | 4 | Qwen2.5-VL | 500 | 87.4 |
| 8 | 4 | Qwen2.5-VL | 500 | 83.9 |

> **Finding 6.** GRPO performance varies significantly with group and batch size configurations, highlighting the importance of empirical hyperparameter tuning.

### 3.7 HOW TO INVOLVE IMAGE RESOLUTION INFORMATION IN THE PROMPT?

Prompting image resolution may provide additionally helpful context, especially for high-resolution GUI screenshots. We explore when such information should be incorporated into the prompt. Specifically, we compare three strategies: (1) never provide resolution; (2) provide resolution during both training and testing; (3) provide resolution only at test time. When resolution is included, we prepend the prompt with *"The screenshot resolution is {width}×{height}."*

Table 7: Effect of image resolution in the prompt under SRF, point prediction, In-Bbox reward, LoRA, $groupsize = 6$, and $batchsize = 4$ on ScreenSpot (Qwen2.5-VL, 500 training steps).

| Train | Test | Backbone | Step | Acc (%) |
|---|---|---|---|---|
| ✓ | ✓ | Qwen2.5-VL | 500 | 83.7 |
| ✗ | ✗ | Qwen2.5-VL | 500 | 87.4 |
| ✗ | ✓ | Qwen2.5-VL | 500 | 88.0 |

As shown in table 7, the highest accuracy is achieved when resolution information is excluded during training but added at test time. We hypothesize that withholding resolution during training may challenge the model to learn a better spatial reasoning ability. At test time, the additional resolution context then serves as a useful signal to refine predictions.

> **Finding 7.** Withholding the cue of image resolution during training fosters better learning, while providing it at test time proves beneficial.

### 3.8 FINAL DESIGN CHOICES FOR RFT ON GUI-VG

Based on the studies above, we finalize a set of design choices for an effective and efficient RFT pipeline for GUI visual grounding under GRPO. We propose the Soft Reward Function (SRF) to provide partial credit for format compliance while relaxing output constraints. For the prediction format, we use direct point prediction with the In-Bbox binary reward. To stabilize training, we introduce the Adversarial KL Factor with a coefficient of $\beta = 1 \times 10^{-4}$. We employ LoRA for efficient fine-tuning and set the group size to 6 and batch size to 4. Image resolution information is withheld during training and added only at inference. We train 1,300 steps for our final version.

## 4 COMPARISON WITH PREIVOUS METHODS

We compare our final RFT method against prior approaches across three GUI-VG benchmarks introduced in section 2.2: ScreenSpot [Cheng et al., 2024], ScreenSpot v2 [Wu et al., 2024], and ScreenSpot-Pro [Li et al., 2025].

Table 8: Comparison of various models on ScreenSpot. The optimal result is **bolded**. "Size" refers to model size. "#Train" refers to training samples.

| Method | Size | #Train | Mobile Text | Mobile Icon | Desktop Text | Desktop Icon | Web Text | Web Icon | Avg. |
|---|---|---|---|---|---|---|---|---|---|
| Fuyu [Bavishi et al., 2023] | 8B | – | 41.0 | 1.3 | 33.0 | 3.6 | 33.9 | 4.4 | 19.5 |
| CogAgent [Hong et al., 2023] | 18B | 400K | 67.0 | 24.0 | 74.2 | 20.0 | 70.4 | 28.6 | 47.4 |
| SeeClick [Cheng et al., 2024] | 9.6B | 364K | 78.0 | 52.0 | 72.2 | 30.0 | 55.7 | 32.5 | 53.4 |
| AGG [Lei et al.] | 0.4B | 35M | 86.1 | 62.8 | 81.8 | 46.2 | 74.2 | 48.4 | 66.6 |
| OmniParser [Lu et al., 2024] | * | – | 93.9 | 57.0 | 91.3 | 63.6 | 81.3 | 51.0 | 73.0 |
| UGround [Gou et al., 2024] | 7B | 1.3M | 82.8 | 60.3 | 82.5 | 63.6 | 80.4 | 70.4 | 73.3 |
| ShowUI-G [Lin et al., 2024] | 2B | 119K | 91.6 | 69.0 | 81.8 | 59.0 | 83.0 | 65.5 | 74.9 |
| ShowUI [Lin et al., 2024] | 2B | 256K | 92.3 | 75.5 | 76.3 | 61.1 | 81.7 | 63.6 | 75.1 |
| OS-Atlas [Wu et al., 2024] | 4B | 13.58M | 85.7 | 58.5 | 72.2 | 45.7 | 82.6 | 63.1 | 68.0 |
| OS-Atlas [Wu et al., 2024] | 7B | 13.58M | 93.0 | 72.9 | 91.8 | 62.9 | 90.9 | 74.3 | 81.0 |
| GuirlVG | 7B | 2K | 96.3 | 86.0 | 93.3 | 77.1 | 91.7 | 83.5 | 88.0 |
| GuirlVG | 7B | 5.2K | 96.0 | 84.7 | 92.8 | 80.0 | 92.6 | 85.9 | **88.7** |

Results on the ScreenSpot benchmark are shown in table 8. Our method substantially outperforms previous methods that rely on supervised fine-tuning (SFT), despite using significantly fewer training samples. Specifically, while prior SFT methods are trained on hundreds of thousands to over ten million examples, our RFT method achieves superior performance with just 2K training samples. For example, we outperform OS-Atlas—*which uses 6.79K times more data*—by **+7.0%** in accuracy, highlighting the efficiency and effectiveness of RFT as a post-training strategy. When increasing training to 1300 steps using 5.2K training samples, our method achieves further improvements, outperforming OS-Atlas by **+7.7%**. Notably, on the Mobile-Icon subset, our method exceeds OS-Atlas by **+11.8%**, despite our training data containing no mobile-specific samples. This suggests that RFT enhances out-of-domain reasoning capabilities, aligning with the claim from Chu et al. [2025] that "SFT memorizes, RL generalizes."

Table 9: Comparison of various models on ScreenSpot v2. The optimal result is **bolded**. "Size" refers to model size. "#Train" refers to training samples.

| Method | Size | #Train | Mobile Text | Mobile Icon | Desktop Text | Desktop Icon | Web Text | Web Icon | Avg. |
|---|---|---|---|---|---|---|---|---|---|
| SeeClick [Cheng et al., 2024] | 9.6B | 364K | 78.4 | 50.7 | 70.1 | 29.3 | 55.2 | 32.5 | 55.1 |
| OS-Atlas [Wu et al., 2024] | 4B | 13.58M | 87.2 | 59.7 | 72.7 | 46.4 | 85.9 | 63.1 | 71.9 |
| OS-Atlas [Wu et al., 2024] | 7B | 13.58M | 95.2 | 75.8 | 90.7 | 63.6 | 90.6 | 77.3 | 84.1 |
| GuirlVG | 7B | 2K | 99.3 | 89.6 | 94.8 | 72.9 | 95.7 | 83.3 | 90.9 |
| GuirlVG | 7B | 5.2K | 98.3 | 89.6 | 94.3 | 80.7 | 95.7 | 86.2 | **91.9** |

Results on ScreenSpot v2 (table 9) mirror the trends observed on ScreenSpot. With only 2K training examples, our method surpasses all previous methods, and with 5.2K examples, it reaches a new state-of-the-art of **91.9%** average accuracy—**+7.8%** higher than OS-Atlas (7B). Performance gains are consistent across all subdomains, reaffirming the generalization strength of our RFT pipeline.

Table 10: Comparison of various models on ScreenSpot-Pro. The optimal result is **bolded**.

| Model | Development Text | Development Icon | Creative Text | Creative Icon | CAD Text | CAD Icon | Scientific Text | Scientific Icon | Office Text | Office Icon | OS Text | OS Icon | Avg |
|---|---|---|---|---|---|---|---|---|---|---|---|---|---|
| SeeClick [Cheng et al., 2024] | 0.6 | 0.0 | 1.0 | 0.0 | 2.5 | 0.0 | 3.5 | 0.0 | 1.1 | 0.0 | 2.8 | 0.0 | 1.1 |
| OS-Atlas-4B [Wu et al., 2024] | 7.1 | 0.0 | 3.0 | 1.4 | 2.0 | 0.0 | 9.0 | 5.5 | 5.1 | 3.8 | 5.6 | 0.0 | 3.7 |
| ShowUI-2B [Lin et al., 2024] | 16.9 | 1.4 | 9.1 | 0.0 | 2.5 | 0.0 | 13.2 | 7.3 | 15.3 | 7.5 | 10.3 | 2.2 | 7.7 |
| CogAgent-18B [Hong et al., 2023] | 14.9 | 0.7 | 9.6 | 0.0 | 7.1 | 3.1 | 22.2 | 1.8 | 13.0 | 0.0 | 5.6 | 0.0 | 7.7 |
| Aria-GUI [Yang et al., 2024] | 16.2 | 0.0 | 23.7 | 2.1 | 7.6 | 1.6 | 27.1 | 6.4 | 20.3 | 1.9 | 4.7 | 0.0 | 11.3 |
| UGround-7B [Gou et al., 2024] | 26.6 | 2.1 | 27.3 | 2.8 | 14.2 | 1.6 | 31.9 | 2.7 | 31.6 | 11.3 | 17.8 | 0.0 | 16.5 |
| OS-Atlas-7B [Wu et al., 2024] | 33.1 | 1.4 | 28.8 | 2.8 | 12.2 | 4.7 | 37.5 | 7.3 | 33.9 | 5.7 | 27.1 | 4.5 | 18.9 |
| GuirlVG-2K-7B | 57.8 | 9.0 | 38.9 | 10.5 | 26.9 | 7.8 | 44.4 | 14.5 | 57.1 | 22.6 | 39.3 | 14.6 | 31.6 |
| GuirlVG-5.2K-7B | 64.9 | 7.6 | 42.9 | 11.2 | 28.9 | 9.4 | 63.9 | 16.4 | 63.8 | 26.4 | 43.9 | 13.5 | **36.1** |

Finally, results on ScreenSpot-Pro (table 10) demonstrate the strong generalization of our method to high-resolution, professional GUIs. With only 2K training examples, our approach already

outperforms all prior methods by a large margin, achieving **31.6%** average accuracy—surpassing the best SFT baseline OS-Atlas (7B) by **+12.7%**. Scaling up to 5.2K examples further boosts performance to **36.1%**, an absolute gain of **+17.2%** over OS-Atlas. This trend is consistent across all domains, including particularly challenging ones like Creative, CAD and OS, confirming the robustness of our RFT pipeline in complex real-world scenarios.

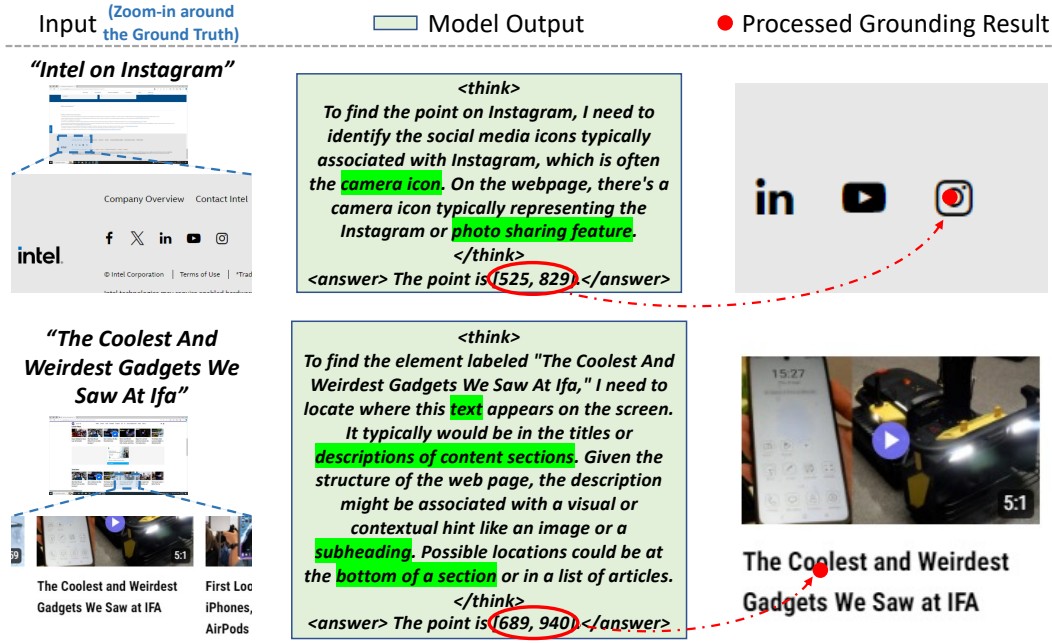

Figure 2: Qualitative Result of *GuirlVG*.

## 5 QUALITATIVE RESULTS

In this section, we present qualitative results to illustrate the reasoning capabilities of *GuirlVG* in GUI visual grounding tasks. fig. 2 shows two representative examples, each consisting of the input (left), model output with intermediate thinking steps (middle), and the final grounding result (right). The thinking process is highlighted with green color. In the first example, the task is to locate an icon on a webpage. *GuirlVG* begins by reasoning that it needs to identify social media icons, often represented by a camera icon. Recognizing the webpage context, the model correctly identifies the Instagram icon and grounds the instruction to the coordinates, as shown in the red dot in the grounding result. In the second example, the instruction is to find a text. *GuirlVG* reasons that the target is a text and it is likely to appear as a contextual hint like a subheading. By analyzing the structure of the webpage, the model further reasons that the target is at the bottom of a section. These qualitative results underscore *GuirlVG*'s textual understanding and advanced reasoning abilities, enabled by our reinforcement learning-based approach. By explicitly modeling the thinking process, *GuirlVG* not only achieves high accuracy but also provides interpretable steps, making it a reliable solution for GUI-VG tasks.

## 6 CONCLUSION

In this work, we revisit the paradigm of post-training for GUI visual grounding and present the first comprehensive empirical study of rule-based reinforcement fine-tuning (RFT) in this domain. Through systematic analysis and a series of targeted innovations—including the decomposition of GRPO components, introduction of the Adversarial KL Factor, and extensive tuning of training configurations—we demonstrate that RFT, when properly optimized, decisively outperforms supervised fine-tuning (SFT). Using as few as 2K training examples, our method surpasses strong SFT baselines trained on orders of magnitude more data across three challenging benchmarks, achieving new state-of-the-art performance. These findings challenge the prevailing reliance on large-scale SFT and highlight RFT as a more data-efficient and generalizable alternative for GUI-VG.

## ACKNOWLEDGEMENT

This research is supported by NSF IIS-2525840, CNS-2432534, ECCS-2514574, NIH 1RF1MH133764-01 and Cisco Research unrestricted gift. This article solely reflects opinions and conclusions of authors and not funding agencies.

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

## A APPENDIX

### A.1 ADDITIONAL DETAILS FOR SECTION 3.1

We provide additional details of the trivial adoption of RFT (RFT-trivial) and the implementation of SFT, which contributes to the reproducibility of the results of this paper.

---
**Algorithm 1** Format Reward Calculation

---
1: **function** FORMATREWARD(completion)
2:      $pattern \leftarrow$ regex `"<think>.*?</think>\s*<answer>.*?</answer>"`
3:      **return** 1.0 if completion matches $pattern$ else 0.0
4: **end function**

---

---
**Algorithm 2** Accuracy Reward Calculation

---
1: **function** ACCURACYREWARD(completion, GT_box)
2:      $answer\_pattern \leftarrow$ regex `<answer>(.*?)</answer>`
3:      $bbox\_pattern \leftarrow$ regex `[(\d+),\s*(\d+),\s*(\d+),\s*(\d+)]`
4:      $reward \leftarrow 0.0$
5:      **if** completion matches $answer\_pattern$ **then**
6:          $pred\_bbox \leftarrow$ find the match in completion
7:          **if** length of $pred\_bbox$ is 4 **and** IoU($pred\_bbox$, GT_box) $> 0.5$ **then**
8:              $reward \leftarrow 1.0$
9:          **end if**
10:      **end if**
11:      **return** $reward$
12: **end function**

---

The format reward function of RFT-trivial is shown in algorithm 1 and the corresponding accuracy reward function is shown in algorithm 2.

For the SFT baseline, we use the following prompt:

*Please provide the bounding box coordinates of the region described by this sentence: <description>.*

The answer format is:

*json \n [bbox_2d: <ground-truth bounding box>, label: <description>] \n.*

We adopt the official evaluation code of Qwen2.5-VL[1] to obtain the zero-shot baseline performance.

### A.2 ADDITIONAL DETAILS FOR SECTION 3.2

We provide details of our Soft Format Reward in algorithm 3 to help readers better understand it.

The prompt we use in section 3.2 is as follow:

*Please provide the bounding box coordinates [x1, y1, x2, y2] of a specific element based on this sentence: <description>. First, think about the reasoning process in the mind within <think> </think> tags. Then, output the bounding box coordinates within <answer> </answer> tags.*

### A.3 MATHEMATICAL FORMULATION OF REWARD FUNCTIONS

We provide the formal mathematical definitions for the reward functions and design choices discussed in the main methodology sections.

---

[1] https://github.com/QwenLM/Qwen2.5-VL/blob/main/cookbooks/computer_use.ipynb

---

**Algorithm 3** Soft Format Reward Calculation

---

1: **function** SOFTFORMATREWARD(completion)
2:     $score \leftarrow 0$
3:     **if** "<think>" in completion **then**
4:         $score \leftarrow score + 0.5$
5:     **end if**
6:     **if** "< /think>" in completion **then**
7:         $score \leftarrow score + 0.5$
8:     **end if**
9:     **if** full "<answer>...< /answer>" block detected **then**
10:         $score \leftarrow score + 2/3$
11:         **if** exactly two numbers found inside the block **then**
12:             $score \leftarrow score + 1/3$
13:         **end if**
14:     **else if** "<answer>" or "< /answer>" detected **then**
15:         $score \leftarrow score + 1/3$
16:     **end if**
17:     **return** $score/2$                 ▷ normalized by the maximum possible reward
18: **end function**

---

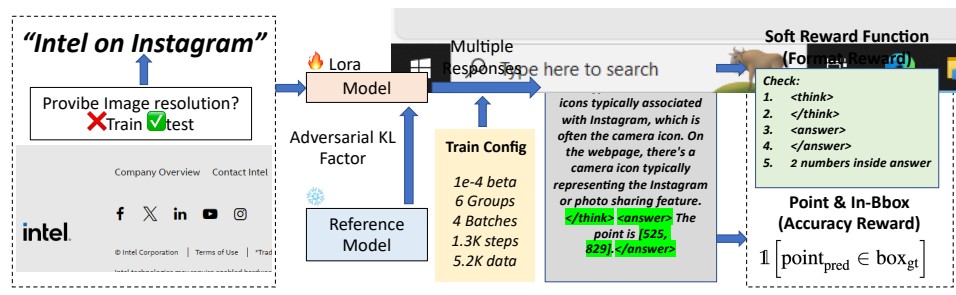

Figure 3: The overview of GuirlVG.

### A.3.1   FORMULATION FOR SECTION 3.1 (TRIVIAL RFT)

In the trivial RFT setting, rewards are strictly coupled to exact string matching. Let $\mathbb{1}[\cdot]$ denote the indicator function.

$$R_{\text{format}} = \mathbb{1}[\text{match } \langle think \rangle ... \langle /think \rangle ... \langle answer \rangle ... \langle /answer \rangle]$$
$$R_{\text{acc}} = \mathbb{1}[\text{match } \langle answer \rangle ... \langle /answer \rangle] \cdot \mathbb{1}[\text{match bbox array with 4 numbers}] \cdot$$
$$\mathbb{1}[\text{IoU}(\text{box}_{\text{pred}}, \text{box}_{\text{gt}}) > 0.5]$$

### A.3.2   FORMULATION FOR SECTION 3.2 (SOFT REWARD FUNCTION)

To mitigate the sparse reward signal caused by strict syntax constraints, the Soft Reward Function (SRF) decomposes the format reward into partial credits, relaxing output style:

$$R_{\text{format}} = \frac{1}{2} \left( \frac{1}{2} \mathbb{1}[\text{match } \langle think \rangle] + \frac{1}{2} \mathbb{1}[\text{match } \langle /think \rangle] + \frac{1}{3} \mathbb{1}[\text{match } \langle answer \rangle] \right.$$
$$\left. + \frac{1}{3} \mathbb{1}[\text{match } \langle /answer \rangle] + \frac{1}{3} \mathbb{1}[\text{match 4 numbers inside answer}] \right)$$

### A.3.3   FORMULATION FOR SECTION 3.3 (ACCURACY REWARD VARIANTS)

We empirically compare different designs for model prediction combined with the corresponding accuracy reward function. The formulations for the variants are defined as follows:

$$
\begin{aligned}
R_{\text{IoU@0.5}} &= \mathbb{1}\big[\text{IoU}(\text{box}_{\text{pred}}, \text{box}_{\text{gt}}) > 0.5\big] \\
R_{\text{IoU}} &= \text{IoU}(\text{box}_{\text{pred}}, \text{box}_{\text{gt}}) \\
R_{\text{Distance@k}} &= \mathbb{1}\Big[\|\text{point}_{\text{pred}} - \text{point}_{\text{gt}}\|_2 \leq 80\Big] \\
R_{\text{In-Bbox}} &= \mathbb{1}\Big[\text{point}_{\text{pred}} \in \text{box}_{\text{gt}}\Big]
\end{aligned}
$$

### A.3.4 FORMULATION FOR OTHER SECTIONS

For Section 3.4, the formulation for the Adversarial KL Factor is provided in Eq. 3.1 of the main paper. For Sections 3.5, 3.6, and 3.7, the underlying optimization equation remains the GRPO, consistent with Eqs. 2.1, 2.2, and 2.3 provided in the Preliminaries section.

### A.4 OVERVIEW OF GUIRLVG

As shown in fig. 3, we provide the overview of GuirlVG. Based on our empirical results, we finalize a set of design choices for GUI visual grounding under GRPO. We propose the Soft Reward Function (SRF) to provide partial credit for format compliance while relaxing output constraints. For the prediction format, we use direct point prediction with the In-Bbox binary reward. To stabilize training, we introduce the Adversarial KL Factor with a coefficient of $\beta = 1 \times 10^{-4}$. We employ LoRA for efficient fine-tuning and set the group size to 6 and batch size to 4. Image resolution information is withheld during training and added only at inference. We train 1,300 steps for our final version.

