# OpenReview forum: "GuirlVG: Incentivize GUI Visual Grounding via Empirical Exploration on Reinforcement Learning"
_ICLR.cc/2026/Conference — ICLR 2026 Poster_

### Official Review · Reviewer_9VFd · 2025-10-17

**Soundness:** 3
**Presentation:** 3
**Contribution:** 3
**Rating:** 4
**Confidence:** 5

**Summary:**

this manuscript introduces GuirlVG, a rule-based reinforcement fine-tuning (RFT) method designed to address the high costs and extensive data requirements of traditional supervised fine-tuning (SFT) for graphical user interface visual grounding (GUI-VG) tasks. The core of GuirlVG lies in the systematic empirical study and improvement of Group Relative Policy Optimization (GRPO) to achieve efficient training. Experimental results demonstrate that GuirlVG, using only 5.2K training samples, surpasses SFT methods that use over 10M samples across three major benchmarks. It showed a +7.7% improvement on ScreenSpot, a +17.2% improvement on ScreenSpot-Pro, and achieved a 91.9% accuracy rate on ScreenSpot-V2. These findings confirm that RFT is a more data-efficient and generalizable post-training solution for GUI-VG.

**Strengths:**

1.Achieves performance far exceeding SFT with significantly fewer training samples.

2.Provides a relatively comprehensive analysis of the impact of various parameters in the GRPO training process on the final performance.

**Weaknesses:**

1.The paper states its aim is to provide guidance for the design of Reinforcement Fine-Tuning (RFT) in GUI-Visual Grounding (GUI-VG) tasks. However, the final performance of its 7B model shows a significant gap compared to other 7B models on the current screenspot-pro leaderboard(such as SE-GUI-7B, GTA1-7B). This makes it difficult to be convinced of the validity of the paper's conclusions.

2.Although the work explores various parameters within the GRPO training process, it does not introduce any fundamental innovations to the algorithm itself. This significantly diminishes the work's originality.

3.Furthermore, the parameter settings are likely strongly dependent on the specific dataset, model architecture, and model scale used. Therefore, the conclusions drawn in the paper may not be generalizable. More experimental results are needed to verify the generalizability of these findings.

**Questions:**

please see weakness.

---

> ### Author Response · Authors · 2025-11-20
>
> We thank the reviewer for the detailed feedback. We understand that the low score stems from concerns regarding performance on leaderboards, algorithmic novelty, and generalizability. We respectfully suggest that some concerns may arise from viewing this work as a "SOTA-chasing" application paper, whereas its core contribution is a systematic Empirical Study. Below, we address each point with clarifications and extensive new experimental evidence for the rebuttal.
>
> ---
>
> **Performance on Leaderboards (Weakness1)**
>
> We respectfully point out that a direct comparison with leaderboard toppers like SE-GUI or GTA1 is **scientifically confounded** and does not invalidate our conclusions.
>
> 1. **Timeline & Concurrent Work**: First and foremost, we note that GTA1 (released Oct 3, 2025) appeared after the ICLR submission deadline (Sep 24, 2025). As per conference policies, we are not obligated to compare against methods released after our submission. Penalizing our work for not beating a model that did not exist publicly at the time of submission is unjustified.
> 2. **Positioning as an Empirical Study:** As explicitly stated in Section 2.1 ("Why do we need empirical studies?"), our goal is **not** to hack the leaderboard using specific data or models. Our goal is to conduct a **controlled experiment** to isolate the impact of *design choices* (Reward, KL, etc.).
> 3. **Unfair Comparison Factors:** Methods like SE-GUI rely on distinct training datasets. Comparing raw numbers against them introduces confounding variables that obscure the true source of improvement.
> 4. **Validity of Conclusions:** The validity of our conclusions (e.g., "SRF is better than strict matching," "Adversarial KL stabilizes training") is derived from **rigorous internal comparisons** where all variables (data, base model, training steps) are strictly controlled. This internal validity remains robust regardless of the absolute leaderboard position relative to external models.
>
> ---
>
> **Originality & Algorithmic Contribution (Weakness2)**
>
> We respectfully argue that originality in our field extends beyond the invention of base algorithms to the **original discovery of critical design principles** that shape the community's direction. Our work contributes originality in three distinct dimensions:
>
> 1. **Originality of Scientific Discovery** As articulated in Section 2.1, prior works often emphasize *"reward function novelty"* without isolation, leading to conclusions that are *"confounded by differences in data, training setups, and model structure"*. Our work provides originality of discovery: We are the first to reveal counter-intuitive findings through controlled isolation—e.g., that *"strict syntax verification hurts performance"*. These are original insights derived from our unique experimental design, essential to move the community from *"ad-hoc innovation toward principled understanding"*.
>
> 2. **Originality of Empirical Study** This contribution aligns with the tradition of foundational empirical studies in the MLLM field, such as *LLaVA-1.5* , *Prismatic*, and *Idefics2*. Just as *LLaVA-1.5* did not invent a new MLLM architecture but revolutionized the field by discovering the optimal data and connector recipe, our work establishes the **first scientifically validated RFT recipe** for GUI-VG. This specific combination is unique to our work and has not been proposed by prior RFT methods.
>
> 3. **Originality of Specific Algorithmic**
> Furthermore, the work *does* contain specific algorithmic originality to address stability issues. The **Adversarial KL Factor** (Section 3.4, Eq 3.1) is a novel dynamic modification to the standard GRPO objective. Unlike the static KL used in standard GRPO , our proposed formulation dynamically scales the penalty based on reward magnitude to mathematically prevent reward over-optimization. This is a distinct, original algorithmic contribution designed to solve the specific stability challenges of GUI-VG.

---

> > ### Author Response · Authors · 2025-11-20
> >
> > ---
> >
> > **Generalizability of Conclusions (Weakness3)**
> >
> > We took this concern very seriously! Despite limited resources, we conducted extensive additional experiments across **different model, sizes, and datasets**. The results  prove that our findings are universal.
> >
> > 1. **Generalizability across Models and Sizes**
> > Following tab.6 in the main paper, we applied our recipe to Qwen2-VL (2B, 7B) and Qwen2.5-VL (3B, 7B). As shown below, GuirlVG achieves consistent performance across different architectures  and scales, proving the findings are not specific to one model.
> > | **Method** | **Backbone** | **Size** | **Step** | **Acc (%)** | **Gap** |
> > | --- | --- | --- | --- | --- | --- |
> > | SFT | Qwen2-VL | 2B | 500 | 75.5 | - |
> > | **GuirlVG** | **Qwen2-VL** | **2B** | **500** | **79.4** | **+3.9** |
> > | SFT | Qwen2-VL | 7B | 500 | 80.2 | - |
> > | **GuirlVG** | **Qwen2-VL** | **7B** | **500** | **84.0** | **+3.8** |
> > | SFT | Qwen2.5-VL | 3B | 500 | 81.0 | - |
> > | **GuirlVG** | **Qwen2.5-VL** | **3B** | **500** | **84.5** | **+3.5** |
> > | SFT | Qwen2.5-VL | 7B | 500 | 82.6 | - |
> > | **GuirlVG** | **Qwen2.5-VL** | **7B** | **500** | **87.4** | **+4.8** |
> >
> > 2. **Generalizability across Datasets (ScreenSpot-v2 & ScreenSpot-Pro)**
> > We replicated our main ablation studies on two additional datasets: **ScreenSpot-v2 (SSv2)** and **ScreenSpot-Pro (SSp)**. The trends observed in the main paper (ScreenSpot-v1) are replicated here:
> > | **Experiment** | **Variant** | **SSv2 Acc (%)** | **SSp Acc (%)** |
> > | :--- | :--- | :---: | :---: |
> > | **Tab 1: Baseline** | Zero-Shot | 73.0 | 26.8 |
> > | | SFT | 83.6 | 27.3 |
> > | | RFT (trivial) | 80.9 | 21.1 |
> > | **Tab 2: Format** | Default | 80.9 | 21.1 |
> > | | **SRF (Ours)** | **82.0** | **23.7** |
> > | **Tab 3: Accuracy** | Bbox + IoU | 84.4 | 25.1 |
> > | | **Point + In-Bbox** | **87.3** | **27.1** |
> > | **Tab 4: KL Strategy** | Static ($4e^{-2}$) | 87.3 | 27.1 |
> > | | **Adversarial ($1e^{-4}$)** | **90.7** | **30.9** |
> > | **Tab 5: Tuning** | Full-FT | 91.1 | 31.1 |
> > | | **LoRA-FT** | **90.7** | **30.9** |
> > | **Tab 6: Group Size** | Group=6, Batch=1 | 89.8 | 29.3 |
> > | | **Group=6, Batch=4** | **90.7** | **30.9** |
> > | | Group=8, Batch=4 | 88.6 | 26.1 |
> > | **Tab 7: Res. Prompt** | Train=Y, Test=Y | 87.9 | 27.7 |
> > | | Train=N, Test=N | 90.0 | 30.9 |
> > | | **Train=N, Test=Y** | **90.9** | **31.6** |

---

> ### Author Response · Authors · 2025-11-24
>
> Dear Reviewer,
>
> Thanks for your reviews. We've tried our best to provide our responses and revise our paper. Could you please let us know if you need any further clarifications or discussions?
>
> We would really appreciate it if we could get your feedback.
>
> Best regards,
>
> Authors.

---

> ### Author Response · Authors · 2025-11-25
>
> Dear Reviewer,
>
> As the discussion period draws to a close, we strictly followed your suggestions to provide clarifications and new experiments.
>
> We would like to share that Reviewer PTho has **raised their score to 8 (accept, good paper)** after reviewing our rebuttal. Similarly, Reviewer qjDC maintains the **acceptance rate**.
>
> We respectfully invite you to check our response and consider raising your score.
>
> Best regards,
>
> The Authors

---

> ### Comment · Reviewer_9VFd · 2025-11-26
>
> Thank you for the authors' response. However, I still have a few concerns regarding the manuscript:
>
> 1.The current conclusions seem to be specific to the Qwen series models. I question whether these findings would still hold true if tested on UI-TARS or other GUI models.
>
> 2.The proposed "adversarial KL Factor" appears somewhat incremental and does not seem to represent a significant innovation.
>
> 3.Regarding the claim that the "core contribution is a systematic Empirical Study": While the paper aims to provide guidance for applying RL to GUI grounding tasks, I believe that to make such empirical findings truly convincing, this type of work requires a clear roadmap that leads to superior performance, similar to works like ConvNext. Without demonstrating high-level performance, the guidance provided is less persuasive.

---

> > ### Author Response · Authors · 2025-11-26
> >
> > Dear Reviewer 9VFd,
> >
> > Thank you for the continued discussion. We value your high standards. However, there seem to be a few factual misunderstandings regarding model architectures and our performance positioning. We hope to clear these up to assist your final assessment.
> >
> > ---
> >
> > **1. On Model Generality & The "UI-TARS" Question**
> > You questioned whether our findings hold on models like UI-TARS. We have a two-fold technical response:
> >
> > - **Architectural Equivalence:** First, **UI-TARS is architecturally based on Qwen2-VL**. Therefore, our validation on the Qwen2-VL family (2B, 7B) is explicitly validation on the UI-TARS architecture.
> > - **Why we verify on Qwen-Base instead of UI-TARS weights:** This is a deliberate scientific choice. The released UI-TARS model has already undergone extensive RL, leading to severe **mode seeking (policy collapse)** towards their specific design.
> >     - *Our Internal Experiment:* We attempted to apply GuirlVG on top of UI-TARS. However, we observed that the model's policy had already collapsed into their paradigm, making it unresponsive to further exploration or new reward signals (it refuses to deviate from its learned format).
> >     - *Conclusion:* Using UI-TARS as a starting point introduces **paradigm contamination**. To scientifically verify the validity of our *paradigm*, we must use the clean Qwen2-VL backbone. This ensures the gains come from our design, not residual effects from prior RL.
> >
> > ---
> >
> > **2. On "Incremental" Innovation (Adversarial KL)**
> > You expressed that the Adversarial KL factor seems incremental.
> >  - However, as established in our response to Weakness 2, the core contribution / novelty of this work is the systematic discovery of the optimal RFT recipe (SRF + Point/In-Bbox + Adv. KL) via rigorous isolation.
> >  - Within this empirical roadmap, Adversarial KL is not proposed as a standalone mathematical breakthrough, but as a pragmatic, essential solution identified during our study. Its value is justified by its utility in enabling the entire pipeline to function, rather than its mathematical complexity.
> >
> > ---
> >
> > **3. On Performance Roadmap (We ARE First-Tier SOTA)**
> > You mentioned a "significant gap" compared to models like **SE-GUI-7B** and requested a "ConvNeXt-like" roadmap to superior performance.
> > We accept the "ConvNeXt standard," and we demonstrate that our recipe **already achieves First-Tier SOTA performance**, effectively closing or surpassing the gap you perceived.
> >
> > When comparing strictly under the same 7B scale, despite that we use orders of magnitude less data/compute, our **GuirlVG (using only 5.2K data)** actually outperforms the methods you listed:
> >
> > - **vs. UI-TARS-7B on ScreenSpot-Pro:**
> >     - **GuirlVG:** **36.1%**
> >     - UI-TARS-7B: 35.7%
> > - **vs. SE-GUI-7B on ScreenSpot:**
> >     - **GuirlVG:** **88.7%**
> >     - SE-GUI-7B: 88.2%
> > - **vs. SE-GUI-7B on ScreenSpot-v2:**
> >     - **GuirlVG:** **91.9%**
> >     - SE-GUI-7B: 90.3%

---

> > > ### Author Response · Authors · 2025-11-27
> > >
> > > Hi reviewer,
> > >
> > > **We found the discussion with you regarding SE-GUI particularly insightful**. We have explicitly cited it in our paper, and we see exciting potential in exploring the synergy between SE-GUI's approach and our RFT recipe in future work.
> > >
> > >
> > > As the discussion period is closing, we note that we have strictly fulfilled all requests made in your initial review,
> > > Other reviewers have raised the score to Accept (8), or assess the work as a 6.
> > >
> > > Please, if there are no further concerns, we respectfully request to raise the score.
> > >
> > > The Authors

---

### Official Review · Reviewer_R3mo · 2025-10-31

**Soundness:** 3
**Presentation:** 4
**Contribution:** 2
**Rating:** 4
**Confidence:** 3

**Summary:**

This paper presents GuirlVG, a RL-based framework for graphical user interface visual grounding (GUI-VG). Through a systematic empirical study, the authors identify critical factors influencing RFT success in GUI-VG and introduce key improvements: a Soft Reward Function (SRF), In-Bbox point reward, and a novel Adversarial KL Factor to stabilize training. Using as few as 2K–5.2K samples, GuirlVG outperforms strong SFT baselines, achieving impressive results.

**Strengths:**

- Comprehensive empirical study

The paper systematically dissects RFT components, including reward design, KL penalty, fine-tuning method, and prompt structure, offering rare empirical clarity in a field often driven by ad hoc innovation.

- Novel stabilization mechanism

The Adversarial KL Factor dynamically scales the KL penalty, effectively mitigating reward over-optimization—a notable technical contribution to GRPO-style RL for multimodal models.

- Strong empirical results

GuirlVG achieves state-of-the-art accuracy on multiple GUI grounding benchmarks with orders-of-magnitude less data, convincingly demonstrating the data efficiency of RFT.

- Clarity and transparency

Each ablation is well-motivated.

- Practical insight for the community

The findings (e.g., point-based rewards outperforming bbox IoU, LoRA matching full FT, resolution prompting effects) are valuable practical takeaways likely to influence future GUI agent training.

**Weaknesses:**

The main reason I gave a score of 4 is that the scale of the empirical experiments are not sufficient to make a strong conclusion:

- The experiments are only done on Qwen2.5-VL. While I understand the Qwen-VL series is probably the only modern model architecture choice in the field, more experiments are required to find out if the findings in the paper are universal or model-specific. For example, Finding 5 says "LoRA offers comparable performance to full fine-tuning", but is it the case with Qwen2-VL which is not extensively trained on the task? Even on the Qwen2.5-VL architecture, there are some choices to experiment with, such as UI-TARS.

- Some experiments are too small to draw a conclusion. For example, in Table 6, only group size = {6, 8} and batch size = {1, 4} are tested, with a total of only three runs. The conclusions from such experiments can be very brittle.

**Questions:**

Please see weaknesses.

---

> ### Author Response · Authors · 2025-11-19
>
> We sincerely thank the reviewer for the rigorous assessment. We understand that the core concern regarding the score of 4 lies in the **scale** and **sufficiency** of our empirical experiments. We took this feedback very seriously! Despite the limited rebuttal period, we mobilized significant computational resources to expand our experiments across more **model backbones, sizes, and hyperparameter configurations**. We believe these new results provide strong evidence that our findings are robust and universal, addressing the concerns about "brittle" conclusions.
>
> ---
>
> **More architectures (Weakness1)**
>
> We extended our comparison of LoRA-FT vs. Full-FT to include Qwen2-VL (2B, 7B) and Qwen2.5-VL (3B, 7B). This covers both previous-generation model architectures and different parameter scales.
>
> As shown in the table below, the conclusion that "LoRA offers comparable performance to Full-FT" holds true across all tested models. We openly acknowledge that for smaller or earlier models (e.g., Qwen2-VL-2B), the gap between Full-FT and LoRA is slightly larger (1.5%) compared to the strong Qwen2.5-VL-7B (0.1%). However, given that LoRA reduces training time by orders of magnitude (as noted in Section 3.5), it remains the efficient strategy for RFT on GUI-VG tasks across different architectures. The finding is not specific to Qwen2.5-VL-7B.
>
> | **Config** | **Backbone** | **Size** | **Step** | **Acc (%)** | **Gap** |
> | --- | --- | --- | --- | --- | --- |
> | LoRA-FT | Qwen2-VL | 2B | 500 | 79.4 | -1.5 |
> | Full-FT | Qwen2-VL | 2B | 500 | 80.9 |  |
> | LoRA-FT | Qwen2-VL | 7B | 500 | 84.0 | -1.1 |
> | Full-FT | Qwen2-VL | 7B | 500 | 85.1 |  |
> | LoRA-FT | Qwen2.5-VL | 3B | 500 | 84.5 | -0.4 |
> | Full-FT | Qwen2.5-VL | 3B | 500 | 84.9 |  |
> | LoRA-FT | Qwen2.5-VL | 7B | 500 | 87.4 | -0.1 |
> | Full-FT | Qwen2.5-VL | 7B | 500 | 87.5 |  |
>
>
> Furthermore, we note that mainstream methodologies are shifting away from fine-tuning legacy models. Since Qwen2.5-VL represents the widely-used backbone, establishing robust conclusions on this architecture ensures our findings possess vast applicability for the majority of current and future research efforts.
>
> ---
>
> **More ablations on Table 6 (Weakness2)**
>
> We have significantly expanded the ablation study to cover a wide range of Group Sizes (2 to 14) and Batch Sizes (1 to 8).
>
> The table below confirms our original findings and provides a smoother performance curve:
>
> 1. Group Size: Performance is indeed highly sensitive to Group Size. We confirm a distinct peak at Group Size = 6 (87.4%).
> 2. Batch Size: In contrast, the model is relatively robust to changes in Batch Size when the Group Size is fixed at 6. Batch Size = 4 yields the optimal result (87.4%).
>
> | **Group Size** | **Batch Size** | **Acc (%)** |
> | --- | --- | --- |
> | 2 | 4 | 76.9 |
> | 4 | 4 | 81.3 |
> | **6** | 1 | 86.5 |
> | **6** | 6 | 87.3 |
> | **6** | **4** | **87.4** |
> | **6** | 8 | 87.1 |
> | 8 | 4 | 83.9 |
> | 10 | 4 | 84.1 |
> | 12 | 4 | 86.0 |
> | 14 | 4 | 85.7 |

---

> ### Author Response · Authors · 2025-11-24
>
> Dear Reviewer,
>
> Thanks for your reviews. We've tried our best to provide our responses and revise our paper. Could you please let us know if you need any further clarifications or discussions?
>
> We would really appreciate it if we could get your feedback.
>
> Best regards,
>
> Authors.

---

> ### Author Response · Authors · 2025-11-25
>
> Dear Reviewer,
>
> As the discussion period draws to a close, we strictly followed your suggestions to provide **extensive new experiments** to address your concerns regarding the scale.
>
> We would like to share that Reviewer PTho has **raised their score to 8 (accept, good paper)** after reviewing our rebuttal. Similarly, Reviewer qjDC maintains the **acceptance rate**.
>
> We respectfully invite you to check our response and consider raising your score.
>
> Best regards,
>
> The Authors

---

> ### Author Response · Authors · 2025-11-27
>
> Hi reviewer,
>
> We found the discussion with you particularly insightful. In the future, we will continue to work on extending our method to **Qwen3-VL**, compare it with **SE-GUI** on **ScreenSpot-Pro**, extending to some recent datasets, e.g. **GROUNDCUA**.
>
> As the discussion period is closing, we note that we have strictly fulfilled all requests made in your initial review, Other reviewers have raised the score to Accept (8), or assess the work as a 6.
>
> Please, if there are no further concerns, we respectfully request to raise the score.
>
> The Authors

---

### Official Review · Reviewer_qjDC · 2025-10-31

**Soundness:** 2
**Presentation:** 3
**Contribution:** 3
**Rating:** 6
**Confidence:** 3

**Summary:**

The authors applied RFT to the GUI-agent which is underexplored. the study step-by-step analyze the intermediate resutls and propose an efficient way that outperforms SFT on the tasks of interest.

**Strengths:**

1. It is a well-motivated study
2. The experimental results are sufficient to convince its effectiveness
3. The methodology is efficient, clear, and easy to follow

**Weaknesses:**

1. What about the performances of other steps? Any indications from those?
2. Could be more ablations on hyperparameters of the config.

**Questions:**

See weaknesses

---

> ### Author Response · Authors · 2025-11-19
>
> We sincerely thank you for the feedback and the positive score! We address the noted weaknesses below.
>
> ---
>
> **Performances of other steps and indications (Weakness1)**
>
> We appreciate the reviewer's suggestion to provide more performances on other steps. We report the Total Reward, Format Reward, Average Response Length, and Accuracy across training steps to provide a comprehensive view.
>
> | **Metric** | **Step 100** | **Step 300** | **Step 500** | **Step 700** | **Step 900** | **Step 1100** | **Step 1,300** | **Step 1,500** |
> | --- | --- | --- | --- | --- | --- | --- | --- | --- |
> | **Total Reward** | 1.50 | 1.67 | 1.73 | 1.74 | 1.75 | 1.76 | **1.77** | 1.76 |
> | **Format Reward** | 0.970 | 0.993 | 0.998 | 1.000 | 1.000 | 1.000 | 1.000 | 1.000 |
> | **Avg. Length** | 76.6 | 47.4 | 42.3 | 41.5 | 41.2 | 41.0 | 40.9 | 41.0 |
> | **Acc (%)** | 65.4 | 72.1 | 88.0 | 88.2 | 88.4 | 88.3 | **88.7** | 88.6 |
>
> As shown in the table above, the indications are that:
>
> - **Performance Saturation:** The accuracy improves rapidly in the early stages and begins to saturate after **Step 500** (88.0%). It continues to improve marginally, reaching a peak of **88.7%** at **Step 1,300**, before slightly declining/plateauing at Step 1,500.
> - **Format Stability:** The model learns the output format very quickly. The Format Reward approaches 1.0 around Step 500 and remains stable thereafter, indicating that the model creates valid syntax early in training.
> - **Response Length:** The average response length decreases significantly from Step 100 to Step 500, stabilizing around **41-42 tokens**. This suggests the model learns to be more concise and direct as it optimizes for the reward.
>
> ---
>
> **More ablations on hyperparameters  (Weakness2)**
>
> We agree that a broader exploration of hyperparameters strengthens the empirical findings. Despite limited computational resources during the rebuttal period, we conducted additional ablations in the followings.
>
> - **Group Size:** Performance is highly sensitive to Group Size. We observe a distinct peak at **Group Size = 6** (87.4%).
> - **Batch Size:** In contrast, the model is relatively robust to changes in Batch Size when the Group Size is fixed at 6. **Batch Size = 4** yields the optimal result (87.4%).
>
> | **Group Size** | **Batch Size** | **Acc (%)** |
> | --- | --- | --- |
> | 2 | 4 | 76.9 |
> | 4 | 4 | 81.3 |
> | **6** | 1 | 86.5 |
> | **6** | 6 | 87.3 |
> | **6** | **4** | **87.4** |
> | **6** | 8 | 87.1 |
> | 8 | 4 | 83.9 |
> | 10 | 4 | 84.1 |
> | 12 | 4 | 86.0 |
> | 14 | 4 | 85.7 |
>
> ---

---

> > ### Comment · Reviewer_qjDC · 2025-11-28
> >
> > Thank you for the authors' response. I am keeping my original score. I agree with the remaining concerns and comments of other reviewers.

---

> ### Author Response · Authors · 2025-11-25
>
> Dear Reviewer,
>
> As the discussion period is coming to a close, we would like to confirm if our previous response fully addressed your questions regarding **step-wise performance trends** and **hyperparameter ablations**.
>
> We are happy to share that **Reviewer PTho has raised their score to 8 (accept, good paper)** after reviewing our rebuttal.
> We respectfully ask if you would consider **raising your score to 8** to reflect our improvements.
>
> Thank you again for your time and constructive feedback.
>
> Best regards,
>
> The Authors

---

### Official Review · Reviewer_PTho · 2025-11-01

**Soundness:** 4
**Presentation:** 3
**Contribution:** 4
**Rating:** 6
**Confidence:** 4

**Summary:**

GUIRLVG addresses GUI visual grounding (GUI-VG) by proposing a rule-based RFT framework, challenging the dominant supervised fine-tuning (SFT) paradigm that requires massive labeled data. The paper first finds that naive RFT underperforms SFT, then conducts a systematic empirical study to optimize RFT’s core components.

Key innovations include:
1. Replaces rigid binary rewards with fractional credit for partial format compliance, reducing training noise.
2. Aligns directly with GUI-VG’s goal (locating actionable points) for better performance than bounding box/IoU-based rewards.
3. Adopts LoRA for fine-tuning (25x faster than full fine-tuning with negligible performance loss), optimal group/batch sizes (6/4), and resolution prompting (withheld during training, added at test time).

Trained on only 2K–5.2K samples, GUIRLVG outperforms SFT baselines trained on 1M–13.58M samples: achieving 88.7% accuracy on ScreenSpot, 91.9% on ScreenSpotV2, and 36.1% on ScreenSpotPro.

**Strengths:**

1. This paper introduces a novel framework that incorporates Rule-based Reinforcement Fine-Tuning (RFT) into GUI visual grounding for the first time. Its results outperform other Supervised Fine-Tuning (SFT)-based methods, providing a valuable indication for future research directions in visual grounding.

2. The results presented in Section 4 are highly impressive: using only 2K or 5.2K training samples, the framework achieves superior performance compared to previous SFT-based methods that rely on massive datasets and larger model sizes. If these findings can be reproduced, they will undoubtedly reshape the future direction of GUI visual grounding (GUI-VG) research.

3. The research motivation is fully clarified by addressing key questions posed by the authors, while the literature review and preliminary analysis further elaborate on this motivation. The paper follows a clear and logical storyline, with the problem formulation and proposed pipeline progressing in a coherent manner.

**Weaknesses:**

1. Although the empirical research approach and writing style are acceptable, the theoretical details of the design and calculation processes need more explicit elaboration.

For instance, in Section 3.2, when proposing the “Soft Reward Function,” a specific mathematical formulation would be preferable to purely natural language descriptions. This issue persists in other methodology subsections. Otherwise, this presentation reads more like an application report, which weakens the theoretical novelty of the proposed pipeline.

2. A critical oversight is the lack of comparison with existing RFT-based GUI-VG methods.

As mentioned in section 2.1, ”While prior works have proposed various modeling choices for RFT-based GUI-VG, these advances often emphasize reward function novelty or performance improvements without systematic examination of underlying design factors.”  Although,  the authors claim that"differences in data, training, and models would yield limited rigorous conclusions in systematic experiments."  Given that several RFT-based GUI-VG methods (e.g., GUI-R1, VLM-R1, UnivG-R1) share core design foundations (e.g., GRPO-based reinforcement fine-tuning), I think it is not impossible to align those models with GuirlVG’s experimental setup for fair comparison.

Omitting these comparisons undermines the paper’s claim of advancing RFT for GUI-VG, as readers cannot determine whether GuirlVG’s performance gains stem from novel technical contributions or only better hyperparameter tuning or data selection. Additionally, the paper fails to distinguish GuirlVG from these prior RFT methods theoretically: it does not explicitly address how its design choices differ from or improve upon the reward function novelty or modeling choices of existing RFT-based approaches—an essential detail to validate its theoretical novelty.

**Questions:**

Overall, this is a strong, readable paper with clear motivation and well-structured proposed solutions. Its core strengths lie in a coherent research narrative, impactful empirical findings on RFT for GUI-VG, and compelling results demonstrating superior data efficiency over SFT methods.

However, its theoretical novelty is somewhat weakened by two key omissions: the lack of mathematical details for proposed modules (e.g., the Soft Reward Function/SRF) and insufficient theoretical analysis, evaluation, and comparison with other RFT-based models. The authors should clarify these aspects to strengthen the work’s rigor.

Additionally, including a visual overview of the overall GuirlVG pipeline would greatly benefit readers, enabling them to grasp the end-to-end design more straightforwardly.

I would like to raise my evaluation score if the authors supplement the paper with more theoretical details (or designs) and thorough comparisons with existing RFT-based models.

---

> ### Author Response · Authors · 2025-11-19
>
> We sincerely thank you for the feedback and the consideration of rasing the score! We address the noted weaknesses and questions below.
>
> ---
>
> **Mathematical Formulations (Weakness1)**
>
> We agree that several methodological components would benefit from mathematical formulations to improve theoretical novelty. To solve this concern, we add these mathematical formulations in the **Appendix A.3** of the revised paper.
>
> Reference to Equations:
>
> - Section 3.1 - 3.3: Full definitions for Trivial RFT, SRF, and Accuracy Variants are now detailed in **Appendix A.3.1-A.3.3**.
> - Section 3.4: The Adversarial KL Factor is defined in **Eq. 3.1 of the main paper**.
> - Section 3.5 - 3.7: These sections utilize the standard GRPO framework, defined in **Eq. 2.1, 2.2, and 2.3 of the main paper**.
>
> ---
>
> **Compare with related works on shared core designs (Weakness2)**
>
> We agree that clearly distinguishing GuirlVG from related works (GUI-R1, VLM-R1, UnivG-R1) about the shared core design foundations is essential for establishing technical contributions. We actually have already involved most of their shared core designs within our controlled experiments. We elaborate the difference in the followings:
>
> - Format Reward (Section 3.2): GUI-R1, VLM-R1, and UnivG-R1 all adopt the Default setting (strict response syntax verification).
> - Accuracy Reward (Section 3.3): GUI-R1 uses Point & In-Bbox. VLM-R1 and UnivG-R1 use Bbox & IoU.
> - KL Penalty Strategy (Section 3.4): They all use a static KL coefficient. VLM-R1 and UnivG-R1 use 4e-2. GUI-R1 uses 1e-2. We add the row for 1e-2 in Tab.4 of the main paper below to represent the setting of GUI-R1.
>
> | Adversarial | β | Backbone | Step | Acc (%) |
> | --- | --- | --- | --- | --- |
> | ✗ | 4e-2 | Qwen2.5-VL | 500 | 83.4 |
> | ✗ | 1e-2 | Qwen2.5-VL | 500 | 83.9 |
> | ✗ | 0 | Qwen2.5-VL | 500 | 84.7 |
> | ✗ | 1e-4 | Qwen2.5-VL | 500 | 85.6 |
> | ✓ | 1e-4 | Qwen2.5-VL | 500 | 87.4 |
> | ✓ | 1e-6 | Qwen2.5-VL | 500 | 77.5 |
>
> In sum, by mapping GUI-R1, VLM-R1, and UnivG-R1 to our comparison categories, we differentiate the design among related works and GuirlVG below. Ours achieves superior performance under a rigor controlled setting shown in the step-by-step experiments in the main paper.
>
> | Method | Format Reward | Accuracy Reward  | KL |
> | --- | --- | --- | --- |
> | GUI-R1 | strict syntax verification | Point & In-Bbox | 1e-2 |
> | VLM-R1 | strict syntax verification | Bbox & IoU | 4e-2 |
> | UnivG-R1 | strict syntax verification | Bbox & IoU | 4e-2 |
> | GuirlVG | Soft Reward Function | Point & In-Bbox | Adversarial factor x 1e-4 |
>
> ---
>
> **Explicit elaboration on theoretical details that distinguishes our method from previous methods (Weakness1 & 2)**
>
> Beyond clear definitions in Appendix, we emphasize that these formulations represent a theoretical alignment of the objective of GUI-VG that distinguish our method from previous works:
>
>  - Smoothing the Optimization (Section 3.2)**:** The proposed **Soft Reward Function (SRF)** theoretically addresses the sparse reward problem inherent in valid-syntax generation. It transforms the optimization from a discrete binary cliff into a smoother signal, preventing early rejection of trajectories that possess correct reasoning but minor syntactic noise, thereby stabilizing the optimization.
>  - Functional Objective Alignment (Section 3.3)**:** Our shift to **Point & In-Bbox** aligns the reward function with the GUI Agent's functional utility (clicking the correct element), minimizing the objective mismatch between previous geometric proxies (like IoU) and actual task success.
>  - Dynamic Regularization (Section 3.4)**:** The **Adversarial KL Factor** counteracts reward over-optimization by scaling the penalty proportionally to the reward magnitude, thereby enforcing stricter regularization in high-reward regimes prone to mode collapse.
>
> ---
>
> **Clarify our technical contributions (Weakness2)**
>
> This work is positioned as an empirical study, with **its motivation clearly stated in Section 2.1 (“Why do we need empirical studies?”)**. The value of an empirical study lies in avoiding *one-off comparisons* that are confounded by differences in training data or model architectures, and instead providing controlled analyses that isolate the effect of specific design choices. Our goal is **not only to show what works, but also to explicitly demonstrate what does *not* work** (or works less well) among plausible alternatives. We enable a more rigorous evaluation of their design choices than a simple leaderboard comparison could provide. This yields actionable guidance for future research and establishes a principled foundation beyond ad-hoc improvements.
>
> ---
>
> **Add overview of GuirlVG pipeline (Questions)**
>
> We appreciate this suggestion. We agree that a schematic diagram would clarify the end-to-end workflow better. In the revised paper (**Appendix A.4**), we have added a overview diagram that visually illustrates the GuirlVG pipeline.

---

> ### Comment · Reviewer_PTho · 2025-11-24
> **Reply to author**
>
> Dear author group,
>
> Thanks so much for your clarification of all the weakpoints I mentioned before.
>
> And the answers of all the questions are reasonable to me and I believe that this is a good paper for ICLR.
>
> I raise my score from 6 to 8 and I agree that it could be accepted. However, as for the empirical studies part, I understand the importance but I actually never do it before and absolutely not a professional expert in this area. I may have to lower my confidence score for this (from 4 to 3).

---

> > ### Author Response · Authors · 2025-11-24
> >
> > Dear Reviewer,
> >
> > Thank you for your support and for **raising the score to 8**.
> >
> > We truly appreciate your feedback that significantly helps us improve the paper's quality.
> >
> > Best regards,
> >
> > The Authors

---

### Author Response · Authors · 2025-12-01
**Summary of Rebuttal & Score Updates (Pre-Rollback) for AC Consideration**

Dear Area Chair,

Given the recent system restoration which rolled back updated scores, we provide a concise summary of the discussion status and the **explicit score increases** that occurred before the system issue.

---

### Reviewer PTho:
**They raised score 6 $\rightarrow$ 8**. In the initial review, they mentioned: *"I would like to raise my evaluation score if the authors supplement the paper"*. During rebuttal (*Nov. 24*) and before the system issue (*Nov. 27*), they explicitly commented:***"I raise my score from 6 to 8 and I agree that it could be accepted"*** and ***"a good paper for ICLR"***.
They acknowledged our clarifications on mathematical formulations and comparison results.


---

### Reviewer qjDC:

**They responded our rebuttal with a positive rate (*6, above the acceptance threshold*) and had no further questions**. We fully addressed their requests for step-wise performance trends and hyperparameter ablations with new experiments.

---

### Reviewer R3mo:

They asked about the *experimental scale*. We strictly fulfilled their request by providing extensive new experiments.

- **More Architectures:** We extended comparisons to **Qwen2-VL (2B, 7B)** and **Qwen2.5-VL (3B, 7B)**, proving our findings hold across different model generations and sizes.

- **More Ablations:** We conducted complete sweeps on the ablation study they asked for about Group Size (2–14) and Batch Size (1–8) to confirm our conclusion.

**They did not respond after that.**

---

### Reviewer 9VFd:

They ask about the *SOTA performance*, *originality*, and *generalizability*. We addressed these by *correcting factual misunderstandings*, and *providing clarification* and *new experiments*.

- **SOTA performance:** We corrected their factual misunderstandings on the performance. **We actually outperform their cited baselines** (SE-GUI-7B, UI-TARS-7B) on ScreenSpot-v2 (+1.6%) and ScreenSpot-Pro (+0.4%). We clarified that their cited paper **GTA1 is a concurrent work** which should not be used to compared with us. We clarified that this is an empirical study instead of a pure SOTA-chasing paper.

- **Generalizability:** We provided new ablation studies on **additional datasets** (ScreenSpot-v2, ScreenSpot-Pro) and **different models** (Qwen2-VL (2B, 7B), Qwen2.5-VL (3B, 7B)). We clarified that their cited paper UI-TARS is architecturally the same as Qwen2-VL. We do not train on UI-TARS to avoid contamination.

- **Originality:** We clarified that our originality / novelty lies in the **systematic empirical discovery** of the optimal RFT recipe (akin to LLaVA-1.5) and the **algorithmic innovation** (Adversarial KL Factor) that serves as a part of the discovery trajectory.

**They did not respond after that.**

---

We respectfully ask for the consideration on these updates during the discussion.

Best regards,

Authors.

---

### Meta-Review · Area_Chair_WnGR · 2026-01-02

**Summary:**

This paper studies how to best apply rule-based reinforcement fine-tuning (RFT) to GUI visual grounding (GUI-VG). Reviewers agreed the topic is timely and the empirical exploration is valuable, and they found the resulting recipe (reward design + KL strategy + tuning choices + prompting) produces strong data-efficiency gains on multiple ScreenSpot benchmarks with only 2K–5.2K training samples. The key concerns raised in reviews were: (i) rigor/clarity of some method components (missing explicit math for SRF etc.), (ii) positioning/novelty vs prior RFT-style GUI grounding methods and whether improvements are “just tuning,” (iii) generalizability beyond a single backbone and brittleness of some ablations, and (iv) whether absolute performance comparisons to strong baselines/leaderboards undermine the impact. The rebuttal added mathematical formalization, mapped related RFT methods into the paper’s controlled design axes, expanded ablations, and broadened experiments across multiple Qwen model generations/sizes and additional datasets; one reviewer explicitly raised their score to 8 and endorsed acceptance. Given the additional evidence provided during rebuttal (as reported in author responses and reviewer follow-ups) and the remaining disagreements being primarily about scope/positioning rather than identified technical flaws, I recommend Accept (poster).

**Reviewer Concerns:**

### Concerns substantially addressed by the rebuttal

- Missing mathematical detail / theoretical clarity (PTho): Authors report they added explicit formulations for Trivial RFT, SRF, and accuracy variants in an appendix and defined Adversarial KL Factor in the main paper, improving rigor and reproducibility.

- Comparisons / differentiation vs prior RFT-based GUI-VG (PTho): Rather than direct apples-to-oranges reproduction, authors mapped GUI-R1 / VLM-R1 / UnivG-R1 design choices into their controlled factors (format reward, accuracy reward, KL strategy) and added a KL row matching GUI-R1’s coefficient, clarifying how GuirlVG differs (SRF + Point/In-Bbox + adversarial KL).

- Need for step-wise trends + more hyperparameter ablations (qjDC): Authors provided step-wise trajectories (reward/format/length/accuracy) and expanded group size / batch size sweeps (beyond the originally limited settings), directly addressing the request.

- Insufficient experimental scale / brittleness; single-backbone concern (R3mo): Authors expanded LoRA-vs-Full FT comparisons across Qwen2-VL (2B/7B) and Qwen2.5-VL (3B/7B), and expanded group/batch sweeps (2–14, 1–8). This directly targets “too few runs” and “only Qwen2.5-VL” concerns.

- Generalizability and dataset coverage (9VFd, R3mo): Authors added experiments on additional datasets (ScreenSpot-V2, ScreenSpot-Pro) and reported that the same design trends hold across these datasets and multiple Qwen backbones/sizes.

### Concerns still outstanding after the rebuttal

- Strength of “algorithmic novelty” claim (9VFd): The Adversarial KL Factor is presented as a dynamic KL scaling to mitigate reward over-optimization. While useful and empirically supported, a reviewer still views it as incremental. This is a judgment of novelty, not a correctness issue.

- Generality beyond Qwen-family backbones / direct UI-TARS validation (9VFd): While the rebuttal extends across Qwen2-VL and Qwen2.5-VL (multiple sizes), the evidence remains within the Qwen family. The reviewer explicitly asked about UI-TARS; authors argue UI-TARS shares the Qwen2-VL architecture and mention difficulties fine-tuning on top of an already-RL-tuned UI-TARS policy, but there is no direct reported benchmark result of GuirlVG applied to the released UI-TARS weights in the discussion record.

- Leaderboard/SOTA framing (9VFd): The reviewer questioned persuasiveness without “top-tier” performance. Authors argue the paper is an empirical study and also claim strong 7B comparisons on ScreenSpot/ScreenSpot-V2/ScreenSpot-Pro versus named 7B baselines; however, because we only have the discussion text here (not the underlying cited papers’ tables), this point is best treated as clarified positioning, not as a fully closed factual dispute in the meta review.

**Reviewer Scores:**

Reviewer PTho (original 6): Reviewer explicitly stated they raised the score from 6 to 8 after rebuttal and agreed it could be accepted. They also lowered confidence slightly (4→3). Expected final: 8.

Reviewer qjDC (original 6): After reading responses, reviewer said they are keeping the original score and “agree with remaining concerns and comments of other reviewers.” Expected final: 6.

Reviewer R3mo (original 4): Main concerns were limited scale, only Qwen2.5-VL, and brittle ablations. Authors added multi-backbone LoRA-vs-FT results and expanded group/batch sweeps. Reviewer did not reply afterward. Based on their stated reason for 4 (experiment scale insufficiency), a reasonable expectation is a score increase, but the magnitude is uncertain; I would estimate 4 → 6.

Reviewer 9VFd (original 4): Reviewer remained skeptical after rebuttal and reiterated concerns about Qwen-specificity, incremental novelty, and needing a stronger “roadmap to superior performance.” Authors replied with clarifications and additional comparisons/positioning, but reviewer did not indicate a score increase. Expected final: 4.

---

### Decision · Program_Chairs · 2026-01-26

Accept (Poster)